# Children That Are Victims to Civil War: A Social Approach through Art and Culture—Molega, a Traditional Poso Children’s Game: Special Issue

**DOI:** 10.3390/children9070997

**Published:** 2022-07-01

**Authors:** Juraid Abdul Latief, Riady Ibnu Khaldun, Ikhtiar Hatta

**Affiliations:** 1Faculty of Teacher Training and Education, Universitas Tadulako, Jalan Soekarno-Hatta KM.9, Tondo, Mantikulore, Palu City 94231, Indonesia; 2Faculty of Social and Political Sciences, Universitas Sulawesi Barat, Jalan Prof. Dr. Baharuddin Lopa, SH., Baurung, Banggae Timur, Majene,91412, Indonesia; riadyibnukhaldun@rocketmail.com; 3Faculty of Social and Political Sciences, Universitas Tadulako, Jalan Soekarno-Hatta KM.9, Tondo, Mantikulore, Palu City 94213, Indonesia; konsumerisme@yahoo.com

**Keywords:** molega, traditional game, social harmony

## Abstract

Molega is a traditional game that is very popular among children and teenagers and is widely played. Molega has many benefits; it fosters solidarity and social harmony, builds responsibility, and develops honesty, sportsmanship, and egalitarian attitudes and behaviors, which make children happier in terms of mental growth. Before the Poso communal conflict, some games could still be played, especially those that were played in groups. However, several conditions hinder teaching molega to the next generation, such as the increase in modern games that children find more interesting and “prestigious”. In addition, some traditional game tools are difficult to find (such as wood shoots), and the amount of vacant land needed as a game medium continues to decrease. Prolonged conflict has also weakened the molega tradition, but it has had many positive impacts. The results of the social approach applied in this study found that molega, as a children’s game, can foster a spirit of solidarity and social harmony which is relevant and should be to be taught to the current generation of children to create harmony in an increasingly diverse society. Therefore, it inspires a sense of urgency to revitalize various folk games that come from tribes from all regions in Indonesia especially in Central Sulawesi, which can have a long-term positive impact on conflict-affected children. A transformation model requires support from families, schools, and communities to achieve the best results. At a basic education level, special learning modules or studies are needed on the importance of social and community values, which can be introduced by competent institutions at the national level.

## 1. Historical and Social Background

A civil war is a violent conflict or war between two organized groups from the same state or country, which aim to take control of a particular region. Sometimes, this act is also used to achieve independence or a change in government policies for a specific area [1]. A civil war is typically very intense and involves large-scale armed forces, which can result in a high number of casualties and a significant decrease in resources [2]. This war can potentially lead to a decrease in national sovereignty and is usually triggered by foreign intervention and power [3]. There are three prominent causes for a civil war [4]: greed, grievance, and opportunity. Grievances can be the cause of conflicts due to a social cause, such as religion, ethnicity, or status. Opportunity is also a cause of conflict and often comes from a rebellion or the taking over of a national regime. This is a real cause, regardless of whether grievance is used as the main issue to incite people.

The impacts caused by a civil war tend to be more significant than those of a global war due to the limited coverage area, which impacts community life. One of the civil wars that has had a severe long-term impact is the Poso Riot. Poso is a region in Indonesia where civil wars occur regularly. This war started in 1998 and stopped in 2001 via the Malino I agreement; however, civil war continues under the form of a cold war between the parties until the present day. Victims of the conflict tend to live nomadically to protect themselves and their families due to the continuing cold war, which has resulted in a lack of community health and education. The worst impacts are felt by children living in the area. The less-suitable environment causes psychological and emotional problems, such as those experienced by adults. Even though this is common in areas of conflict, a method to overcome this problem is still needed.

A favorable and practical way to overcome human psychological problems is art in its various forms. If artwork is combined with culture, it can create changes in terms of social impact. Because this study focuses on the problems of children as victims of civil war, the approaches and solutions offered must be in the context of their circumstances. In addition to artistic and cultural factors, educational factors must also be a primary consideration when carrying out a social approach that aims to improve the mental condition of children. Poso, found on Celebes Island and in other regions of Asia, has a wealth of cultural heritage in the form of traditional games. With the use of the correct approach, molega as a traditional Poso game can be a suitable solution for these problems. Further testing is needed on the matter, which can then become the basis for a case study. The purpose of this study is to analyze how molega can teach harmony and bring peace to children of the Poso Riot victims and to determine what the impacts of this may be in the future.

## 2. Molega

Molega is a series of games that are played in groups; thus, the game can be considered useful for healing the trauma of the children of Poso Riot victims. Team games can overcome trust issues, as they often involve task division and effective communication [5]. The previous literature has addressed the advantages of these types of activities, such as the development of self-esteem, teaching leadership skills and teamwork, nurturing stronger relationships, better communication, showing respect, and time management [6]. Team games help to develop critical thinking skills, how to accept defeat, building perseverance, controlling emotions, teaching discipline and collaboration, and help parents in supporting their children. As a team game, molega has several rules to fit each style of play [7]. Each practice is explained below.

### 2.1. Gasing (Top) Molega

As the name suggests, this game requires a top as the medium for play. The top is typically a traditional one, made of wood. This game can be played by individuals or groups. The winner is the player with the longest top spin. These games teach abilities, critical thinking, and technical skills to the children who play them, besides that the longer loop will influence the kids emotions because their compete each other to get the longest loop. Figure 1.

### 2.2. Kaniker (Marble) Molega

This game resembles a traditional marble ball game (similar to billiards) played in groups by using vacant land as the arena. One player draws a small circle on the ground to begin the game. All players put a marble in a circle, and then, all the players stand two meters from the ring, behind a line. In turn, they each throw another marble toward the hoop. Players whose marbles are the farthest from the ring can play first. Each player must use marbles outside the circle as “attackers” to hit the marbles in the circle and knock them out of the circle. If a player fails to remove the marbles, the player can take their “attacker” marble to take another turn. The marbles are moved by a flicking motion made by bringing the thumbs together with the middle finger, and then pinching the two fingers on the marbles. “Attacker” marbles must remain in the circle, because the player will lose them if they do not. This game nurtures stronger relationships, critical thinking, respect, and technical skills, while maintaining time management. Figure 2.

### 2.3. Aseng (Fort) Molega

Aseng molega (Figure 3) is a game of traditional fort guard soldiers, played in groups. Instead of using real fortresses, players use lines on the ground and stone as markers, which are often replaced by flowerpots or plants. Before starting the game, the participants make guard lines with chalk like a badminton court; the difference is that there are no double lines at both ends. Then, the players are divided into two teams; each team consists of three to five people or can be adjusted to the total number of participants. One team is a “guard” team, and the other team is an “invader” team. The “guard” team guards the field by keeping the horizontal lines and vertical lines that have been made, whereas the “opponent” team must try to pass each guard line to the back of the line, then go back through the opponent’s guard until they reach the starting line. This game nurtures stronger relationships, critical thinking, respect, communication, and collaboration, while maintaining discipline.

## 3. Approach

### 3.1. Cultural Wisdom in Molega

The concept of community wisdom is a collection of knowledge, actions, and thoughts that are rooted in the culture of one group of humans which is the result of observations over a long period [8]. Culture consists of historically accumulated knowledge, tools, and attitudes that pervade a child’s proximal ecology, including the cultural “practices” of nuclear family members and other kin [9]. Through the socialization process, this wisdom gets passed onto the present generation. These enculturated members of society are themselves subject to a variety of forces in both the natural ecology and society as they carry out their roles, such as caregiving and earning a living. As a result, every aspect of their lives is influenced by these assumptions, values, and beliefs.

Child development is a dynamic and interactive process, because every child has a uniqueness in their interaction with the environments around them. Their mind-shaping comes from what they invoke and receive from others and the environment, which affects how they react and behave [10]. From 0 to 5 years of age, a child experiences massive growth, and biological organization, behavior capabilities, and organizational social experiences are formed [11]. Development requires qualitative changes in a child’s functional organization of their intra-individual brain, body, and behavior, and changes in the relationships between children and their socio-culturally organizational experiences occur [12]. It is evident that children do not make sense of the world consciously and analytically at this age [13].

Molega is a local term in Poso which means fun games among children, forming part of the cultural wisdom. Cultural wisdom in children’s games is seen as social capital to reduce the potential for conflict, and at the same time, create peace and prosperity in society [14]. Various forms of games (object, symbolic and pretend role play) form diverse varieties of cultural environments for learning. However, there are substantial cultural differences in the extent to which adults approve of different forms of games during early childhood [15]. In communities where play is a valued cultural practice for children, children carry out social experimentation with others in play and in daily life [16]. The importance of mutuality and play that transcends the present situation by creating imaginative worlds [17]. Distorting reality in a game paradoxically reinforces learning applied to real life by changing children’s understanding of the relation between objects and meanings [18]. The education taught by molega to its players, of course, will influence the formation of one’s character based on the cultural wisdom contained in the game.

### 3.2. Reharmonization Perspective in Molega

Recognition of the potential for conflict as a dimension for a decrease in harmonization has been expressed by many social scientists, both theoretically and empirically. From various results of research by experts on the conflict in Indonesia there are at least four kinds of sources of conflict in the life of a pluralistic society. A dispute arises if two or more ethnicity groups compete in obtaining a living. Next, a conflict arises if people from a particular ethnicity group impose their cultural beliefs on others [19]. The following reason is if one ethnicity group tries to impose its religious concepts on other groups or tries to dominate others politically. The potential rooted conflict between tribes has been hostile for a long time. In contrast, the root of the civil war in Indonesia is the existence of persistent and accumulative socio-economic disparities covered by ethnic and religious factors [20].

The global economic crisis that had an impact on the deteriorating economic conditions in Indonesia in 1998 triggered national unrest, which made radical groups free to develop. This was exacerbated by the existence of boundaries of settlement patterns by religion-based communities. This process of transition triggered friction between societies, which led to widespread civil war. This type of situation also leads to democratic freedom, which tends to be anarchic. The provocateurs can take advantage of the situation to raise the issue of social inequality caused by problems regarding ethnicity, religion, economy, and injustice caused by the government in power. A decline in public confidence can trigger a civil war, which is what happened in Poso.

Conflicts that occur between adults should not affect social relationships in children, even though the reality is the opposite. Unstable conditions in society lead to restrictions on the interaction of family members with their children, as well as their social sphere. Interaction restrictions can be reduced if parents actively participate in supporting the positive activities of their children, such as full support for the traditional games they play. Molega is believed to be able to have a positive influence on the reharmonization process, as shown when it was implemented in the Watopute district [21]. This study stated the game caused a significant change in mental and behavioral child development, which was reflected in their activities at school. The molega game implemented in the district was initiated by the teachers in the elementary school there. In addition to these changes, the increase in educational values and non-academic achievements also increased by 23%. 

### 3.3. The Value of Social Solidarity and Harmonization in Molega

A game is a voluntary activity which is carried out within certain boundaries of space and time that have been determined according to rules that have been accepted but are fully binding, with a purpose in it, accompanied by feelings of tension and joy and the awareness of daily life [22]. The various forms of games that are played by the community in the Poso Regency, in general, have the spirit of building, teaching, and maintaining harmonization in every relationship and interaction for children. Folk games (such as molega) generally involve many children and automatically strengthen friendships between them. This condition psychologically maintains the strengthening of their social relations during adulthood. Unfortunately, the conflict that occurred in Poso in 1998 has made interest in the game diminish. Friction causes people to not have free time to participate in leisure activities, even those that are very simple, such as playing together with a family member. People, especially adults, prefer to focus on work and other things related to their safety and survival. Molega is a tradition that is increasingly being marginalized. This is getting worse with technological advancements. Parents choose to use technology-based games that further reduce the value of social solidarity and harmonization, which will not be obtained by playing this type of game, which represents the safer choice. Molega has many advantages that are beneficial for a child’s growth [21]. Solidarity is the act of expressing support for other people in the community, or in other words, helping and thinking of others. To express solidarity, a good dose of acceptance, generosity, empathy, and tolerance is needed. All of these points can be obtained and implemented in the molega game. 

## 4. Conclusions

Childhood is a phase of human life that must be passed, but not all children go through the same experiences. The conditions and circumstances in which they grow and develop give different meanings and qualities to that phase and how they go through it. Before the conflict, Poso was a safe and suitable area for children to enjoy their childhood, and children could play with anyone without any restrictions on identity and insecurities. Molega is a type of traditional game that was very popular and was played by many people at that time, from children to teenagers. Molega has an integration of meaning that involves fostering social solidarity and harmonization, building an attitude of responsibility, forming honesty, practicing sportsmanship, and instilling egalitarian attitudes and behaviors that make children happier in their mental growth. Before the conflict, several games could still be taught, especially the types of games carried out in groups. However, several conditions hinder the process of teaching molega to the next generation, such as the incessant attacks from modern games that, according to children, are more attractive and more “prestigious”. In addition, several traditional game tools are challenging to find (such as wood tops), and the availability of vacant land that can be used as a game medium is also decreasing. The prolonged conflict has also caused a weakening of the molega tradition, even though it has many positive impacts.

From the description, depiction, and analysis of this study, it was found that molega fosters social solidarity mechanisms and harmony in the environment for children, which have a positive impact on the development of their psychology in the long run, except when conflicts occur. The results of the social approach applied in this study found that the spirit of solidarity and social harmony that molega fosters means it is still relevant enough to be taught to the children of the present generation to create harmony in this increasingly diverse society. Therefore, it inspires urgency towards the revitalization of various folk games owned by ethnic groups in all regions, which can have a long-term positive impact on conflict-affected children. The transformation model also needs support from families, schools, and communities to get the best results. At the basic education level, a learning module or particular study is needed regarding the importance of social and community values, which can be initiated by the competent institution at the national level.

The limitations of this study are that studies and practical approaches were analyzed through the theory and results of previous reviews, so the results are theoretical. In the next study, a particular study is needed to qualitatively provide results from a child psychology perspective, which will examine the influence of a game on the mental development of children who play a traditional game that brings cultural and social values in an area that is affected by war.

## Figures and Tables

**Figure 1 children-09-00997-f001:**
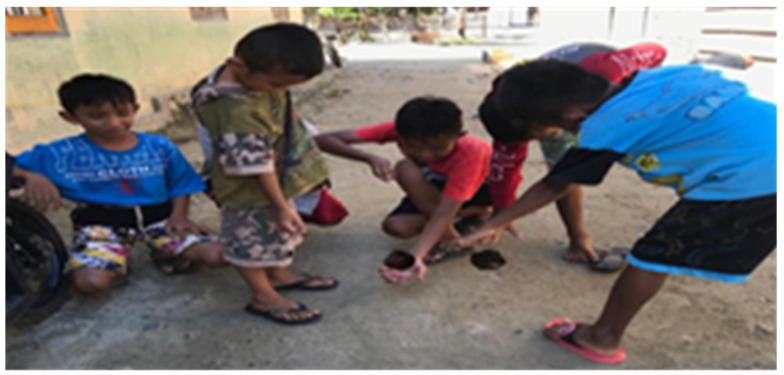
Gasing molega.

**Figure 2 children-09-00997-f002:**
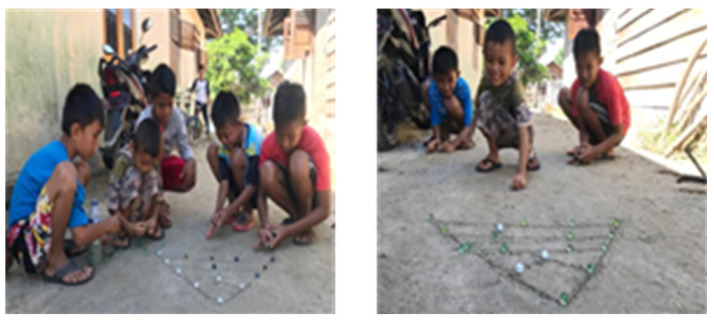
Kaniker molega.

**Figure 3 children-09-00997-f003:**
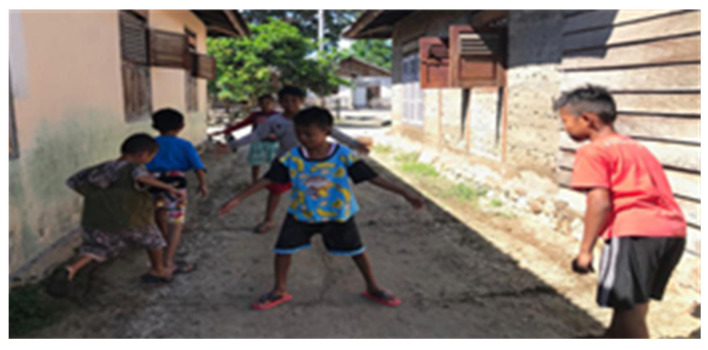
Aseng molega.

## Data Availability

The data presented in this study are available on request from the corresponding author.

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
