# Peer review of "Children That Are Victims to Civil War: A Social Approach through Art and Culture—Molega, a Traditional Poso Children’s Game: Special Issue"

_children, 2022, doi:10.3390/children9070997_

Round 1
Reviewer 1 Report
Dear Author, First of all, thank you for sharing your valuable information. In the study, the valuable traditional game and its meaningful effects are mentioned. The article deals with the game mentioned in its title and the effects of civil wars and conflicts on children's games, but as stated in the research, no research has been conducted investigating the effects of current wars and conflicts on children's games. Although the information disclosed is valuable, presenting the data collected by qualitative or quantitative methods accompanying this information in this article could have made the article much more valuable. I see that this article, which is not a research article, is also not a review article. Because a review article is the summary of other researches and presenting them to the reader as a whole. Literature review is made and as a result of the scanning, a general summary is written for each study and compiled. Unfortunately, this study is not in the category of review article, besides being a research article. In order for the article to be published in journal, I think it is important to present it with the results of the research to be done.
Author Response
Thank you for the comments to our article. We are assuming that our article is already has met the criteria of a research article because the results in this study have examined the relationship between traditional games and conflict victims in Poso.
Reviewer 2 Report
Dear authors,
This is an interesting study and from my reading it shows all elements required to conduct and publish a study have been fulfilled accordingly. I suggest a re-editing for English is necessary before it is publishable. For example, Prolonged conflict in abstract should be replaced with "The prolonged conflict or Prolonged conflicts " in the front of the word.
Thank you
Author Response
Thank you for the comment to our article. We already revising several words and sentences regarding to your instruction.
So for the article that already revised, please see the attachment.

Round 2
Reviewer 1 Report
This article is a poor quality work and It is an article that only informs the reader. I think it will contribute little to the reader from a scientific point of view. For this reason, I think that it is not appropriate to be published in the journal as it is and should be rejected